# Photoprotective Effects of Selected Polyphenols and Antioxidants on Naproxen Photodegradability in the Solid-State

**Kohei Kawabata ***[ID]**, Ayano Miyoshi and Hiroyuki Nishi**

Faculty of Pharmacy, Yasuda Women's University, Yasuhigashi 6-13-1, Asaminami-ku, Hiroshima 731-0153, Japan
* Correspondence: kawabata-k@yasuda-u.ac.jp; Tel.: +81-82-878-9440; Fax: +81-82-878-9540

**Abstract:** Photostabilization is an important methodology to ensure both the quality and quantity of photodegradable pharmaceuticals. The purpose of our study is to develop a photostabilization strategy focused on the addition of photostabilizers. In this study, the protective effects of selected polyphenols and antioxidants on naproxen (NPX) photodegradation in the solid state were evaluated. Residual amounts of NPX were determined by high-performance liquid chromatography (HPLC), and the protective effects of tested additives on NPX photodegradation induced by ultraviolet light (UV) irradiation were evaluated. As a result, quercetin, curcumin, and resveratrol suppressed NPX photodegradation completely. When they were mixed with NPX, the residual amounts of NPX after UV irradiation were significantly higher compared to that without additives, and comparable to those of their control samples. In addition, to clarify the mechanisms of the highly protective effects of these additives on NPX photodegradation, their antioxidative potencies, and UV filtering potencies were determined. There was no correlation between photoprotective effects and antioxidative potencies among selected polyphenols and antioxidants although photoprotective additives showed more significant UV absorption compared to NPX. From these results, it is clarified that a higher UV filtering activity is necessary for a better photostabilizer to photodegradable pharmaceuticals in the solid state.

**Keywords:** naproxen; photodegradation; photoproduct; photoprotective effect; polyphenols; antioxidants; HPLC

## 1. Introduction

Naproxen (NPX, Figure 1A) is a non-steroidal anti-inflammatory drug (NSAID) and chemically (2*S*)-2-(6-Methoxynaphthalen-2-yl)propanoic acid. This pharmaceutical is categorized as 2-aryl propionic acid family including ibuprofen, ketoprofen, and loxoprofen, and utilized for the relief of pains and fevers by the inhibition of cyclooxygenase resulting in the suppression of the generation of prostaglandins. Naixan® tablets, which are the original NPX tablets, have been widely used in clinical situations. It is well known that some NSAID medicines are photosensitive due to their photodegradable active pharmaceutical ingredients (APIs). There are several reports showing the photodegradabilities of NSAIDs such as diclofenac, indomethacin, and sulindac [1–5]. Our previous reports showed that sulindac was photodegraded by both ultraviolet-light (UV) irradiation and sunlight irradiation [5] and converted to *trans*-sulindac by photoisomerization reaction [6]. NPX is also one of the photosensitive pharmaceuticals and photodegraded resulting in the generation of some photo products including 2-acetyl-6-methoxy-naphthalene (Figure 1B), which is the main photoproduct of NPX [7–10]. Amounts of APIs of NPX tablets were much decreased when these tablets were crushed or suspended following UV irradiation [11].

Furthermore, some pharmaceuticals might exert toxicological potencies as a result of the generation of their photoproducts in addition to the loss of beneficial effects derived from the decrease of APIs. For example, 5-diazoimidazole-4-carboxamide is a photoproduct of dacarbazine, which is known as an anti-cancer drug, and induces vascular pain [12]. In

addition, other reports showed the change in biological activities of several pharmaceuticals induced by their photodegradation [5,13–17]. The change of chemical structures, due to the elimination, addition, and rearrangement, might induce changes in biological properties. The photostability of pharmaceuticals is a crucial determinant of their quality and quantity when they are photo-irradiated.

**A. NPX**  **B. NPX photoproduct**  **C. Quercetin**

**Figure 1.** Chemical structures of NPX (**A**), NPX photoproduct (**B**) and quercetin (**C**).

In recent years, photostabilization strategies of photodegradable pharmaceuticals have been developed for their safe use in clinical situations [18]. UV filtering is an efficient method to disrupt photo exposure for APIs. Solar filters and encapsulation have been utilized as a protective barrier to envelop photodegradable pharmaceuticals. Cyclodextrin and its modified forms are major photoprotective careers to perform photostabilization [19,20]. Also, the addition of some antioxidants such as ascorbic acid is one of the photostabilization strategies. Added antioxidants deactivate the excited state of pharmaceuticals, reactive oxygen species, and free radicals generated by UV irradiation, resulting in the exertion of photoprotective effects. Several studies indicated that ascorbic acid attenuates the photodegradation of several photodegradable pharmaceuticals [21–23]. Furthermore, the combination of UV filters and antioxidants showed significant protective effects on the photodegrdation of some pharmaceuticals [21,24]. However, in some cases, the stabilization effect of encapsulation was decreased [25]. These reports suggest that the use of a photostabilizer is limited and the further development of various photostabilization strategies is needed for the photoprotection of lots of photodegradable pharmaceuticals.

In this study, the protective effects of selected polyphenols and antioxidants on NPX photodegradation in the solid state were evaluated. To the best of our knowledge, there are no reports focused on the photostabilization of NPX in the solid state. First, the photoprotective effect of quercetin (Figure 1C) and its dose-dependency were evaluated. Our previous study showed that quercetin, which is one of the antioxidative polyphenols, significantly suppressed NPX photodegradation in an aqueous media [23]. The photoprotective effect of quercetin on NPX photodegradation in the solid state was evaluated. The residual amounts of NPX and the generation rates of NPX photoproducts were estimated utilizing high-performance liquid chromatography (HPLC). Secondly, the photoprotective effects of selected polyphenols and antioxidants (Figure 2) were determined. Finally, both UV absorptive potencies and antioxidative potencies of tested polyphenols were evaluated by means of UV spectral analysis and a test kit for the potential antioxidant (PAO test). The aim of this study is to clarify the efficient photostabilizer for the crushed tablets of a photodegradable pharmaceutical.

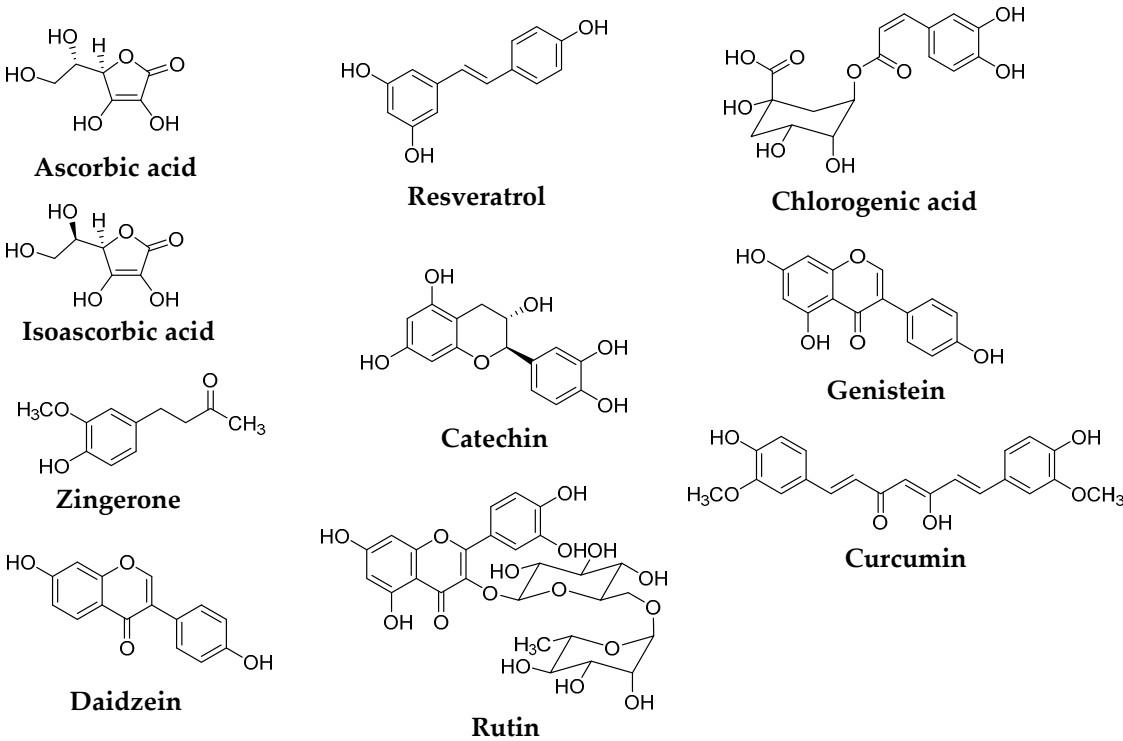

**Figure 2.** Chemical Structures of Selected Polyphenols and Antioxidants.

## 2. Materials and Methods

### 2.1. Materials

NPX, ascorbic acid, isoascorbic acid, zingerone, rutin, daidzein, chlorogenic acid, genistein, resveratrol, methanol, and formic acid were purchased from Fujifilm Wako Pure Chemical Corporation (Osaka, Japan). Catechin, curcumin, and quercetin were purchased from Tokyo Chemical Industry Corporation (Tokyo, Japan). All reagents and organic solvents were of special or HPLC grade. Milli-Q (18.2 Ω/cm) water was prepared by using a Milli-Q water purification system (Merck, Darmstadt, Germany).

### 2.2. Methods

The scheme of the experimental procedure of the evaluation of photostabilization potencies of selected additives for NPX photodegradation is shown in Figure S1 (see Supplementary Materials). Briefly, the difference in residual amounts of NPX in the presence or absence of selected additives after UV irradiation was evaluated.

#### 2.2.1. Preparation of a Test Sample

To determine the dose dependency of quercetin, NPX (10 mmol) and quercetin (1.25 mmol, 2.5 mmol, 5 mmol and 10 mmol) were mixed to make molar ratios for NPX 0.125–1. Also, NPX (10 mmol) and each polyphenol or antioxidant (5 mmol) were mixed to make a molar ratio of NPX 0.5, and these mixtures were used as test samples. 10 mg of test samples were exposed to black light. Control samples were prepared using the same procedures but covered with aluminum foil to interrupt the photo exposure. UV-irradiated samples were dissolved in 100 mL of 50% (*v/v*) methanol and sonicated for 10 min for extraction. The extractions were analyzed by HPLC. All experiments were carried out in quadruplicate.

#### 2.2.2. UV Irradiation Experiment

UV irradiation was carried out in a light cabinet with a black light lamp (20W FL20S BLB, Toshiba, Tokyo, Japan). The most abundant wavelength of this lamp is 365 nm, which is a component of sunlight. UV irradiation intensity at 365 nm was 300 µW/cm²/sec as

measured by a digital radiometer with a 365 nm sensor (UVX-36, UVP, Upland, CA, USA). UV irradiation was carried out at a temperature of 20 °C, and a distance from the lamp source of about 20 cm. Irradiation times were up to 24 h.

### 2.2.3. Evaluation of the Residual Amounts of NPX in UV-Irradiated Samples

The degradation of NPX and the generation of NPX photoproducts were monitored with an HPLC system, which was composed of an LC-20AB pump, a SIL-20AC auto-sampler, an SPD-M20A photodiode array (PDA) detector with LCsolution software, a CBM-20A system controller, a DGU-20A3 degasser, and a CTO-20A column oven (Shimadzu Corp., Kyoto, Japan). Shim-pack Arata C18 column ($4.6 \times 150$ mm, particle size 5 μm, Shimadzu Corp., Kyoto, Japan) was used for HPLC analysis. The column was kept at 40 °C during analysis. The mobile phase was a mixture of methanol and 0.1% formic acid (5:5, *v/v*). Isocratic separations were achieved using this mobile phase. The flow rate was maintained at 1.0 mL/min, and the injection volume was 10 μL. The detection wavelength of NPX and NPX photoproducts was 260 nm. The retention time of NPX was ca. 21 min. Amounts of NPX are shown as the residual rate for amounts of NPX before UV irradiation, calculated by their peak areas.

### 2.2.4. Evaluation of Antioxidative Potencies

The antioxidative potencies of selected polyphenols and antioxidants were evaluated by the PAO test (Nikken SEIL Corp., Shizuoka, Japan). This assay evaluated the antioxidative potencies based on the reduction of $Cu^{2+}$ to $Cu^+$ by tested compounds using the spectrophotometer. The antioxidative potencies of tested compounds were calculated as copper-reducing power (μmol/L). The tested compounds were dissolved in 50% (*v/v*) methanol to make a concentration of 10 mg/L (16.87–60.75 μmol/L) for the PAO test. All experiments were performed in triplicates.

### 2.2.5. UV Spectral Analysis

Tested polyphenols and antioxidants were dissolved in methanol at the final concentration of 10–100 μmol/L. UV absorption spectra were recorded with a V-670 UV/Vis spectrophotometer (JASCO, Tokyo, Japan), interfaced to a PC for data processing. The absorption-maximum wavelength ($\lambda_{max}$, nm) of each compound was obtained from these results. The molar absorption coefficients ($\varepsilon$, L mol$^{-1}$ cm$^{-1}$) were calculated from the absorption of $\lambda_{max}$.

### 2.3. Statistical Analysis

Data are expressed as the mean $\pm$ standard deviation (S.D.). The statistical significance of a difference between two groups was estimated by Student's *t*-test, and between more than three groups was estimated by Tukey's test. The threshold for assessing the significance was $p < 0.05$, $p < 0.01$, or $p < 0.001$.

## 3. Results and Discussion

### 3.1. Dose Dependency of the Photoprotective Effect of Quercetin

In this study, the photoprotective effects of selected polyphenols and antioxidants on the photodegradation of NPX, which is well-known as a photodegradable pharmaceutical [7–10], were evaluated in the solid state. Our previous study indicated that selected polyphenols and antioxidants, such as quercetin, ascorbic acid, and isoascorbic acid, suppressed NPX photodegradation in an aqueous media [23]. Quercetin showed a significant photoprotective effect, so its photoprotective potency for NPX in the solid state was investigated first.

The dose dependency of the protective effect of quercetin on the photodegradability of NPX after UV irradiation was evaluated. HPLC chromatograms of NPX and UV-irradiated NPX with or without quercetin (molar ratio for NPX was 0.5) are shown in Figure 3. The effects of quercetin on residual amounts of NPX and generation rates of NPX photoproducts

are shown in Figure 4. UV irradiation-induced the decrease of NPX peak concomitant with the generation of NPX photoproducts (retention time of the main photoproduct was ca. 18 min, Figure 3B). When a powder of NPX was UV-irradiated without quercetin for 24 h, its residual amount (84.1 ± 3.8%) was significantly less than that of the control sample (101.7 ± 1.1%) as shown in Figure 4A, and the generation rate of the main NPX photoproduct was 4.9 ± 0.7% (Figure 4B). As the same as in our previous study [11], NPX was excited by absorption of the energy derived from UV irradiation, resulting in the elimination of its carboxylic group followed by oxidation for the conversion to the NPX photoproduct in the solid state. Also, both NPX photodegradation and the generation of NPX photoproducts were not observed in the control sample (Figure 3A), showing that other factors such as hydrolysis and temperature did not contribute to the degradation of NPX.

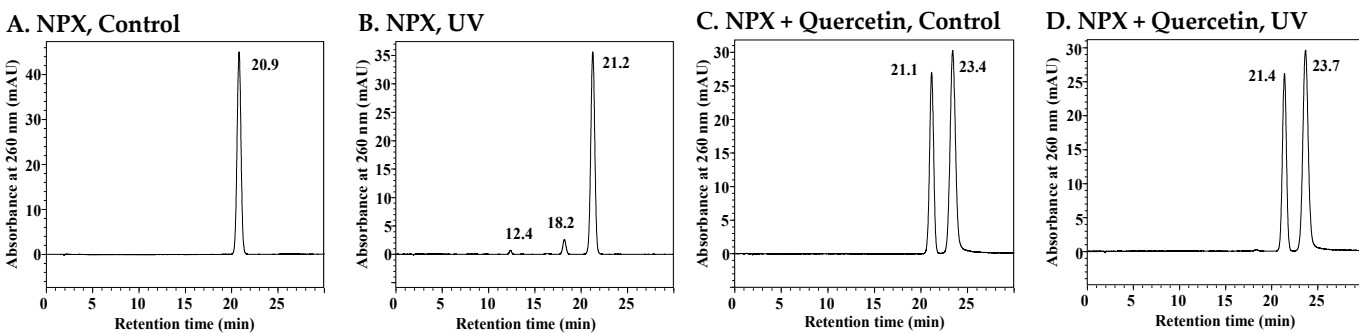

**Figure 3.** HPLC chromatograms of NPX powder and a mixture of NPX and quercetin (molar ratio for NPX is 0.5) with and without UV irradiation for 24 h. (**A**) an NPX powder, (**B**) a UV-irradiated NPX powder, (**C**) a mixture of NPX and quercetin, (**D**) a UV-irradiated mixture of NPX and quercetin. Detection wavelength: 260 nm.

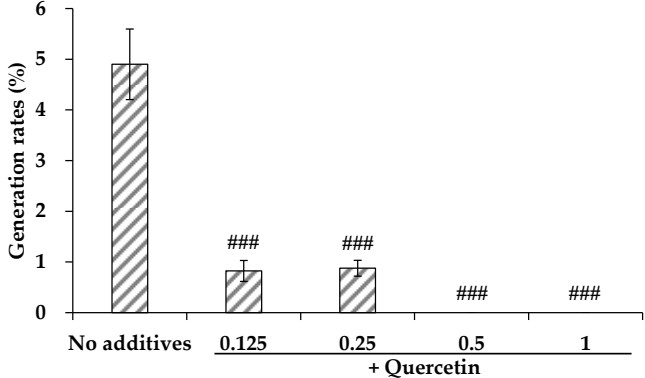

**Figure 4.** Dose dependency of the effect of quercetin (molar ratios for NPX were 0.125–1) on the residual amounts of NPX and the generation rates of the main NPX photoproduct in the UV-irradiated powder. (**A**) Residual amounts of NPX, (**B**) generation rates of the main NPX photoproduct. Values represent mean ± S.D. (n = 4). * Difference compared with control ($p < 0.05$), ** difference compared with control ($p < 0.01$) and ## difference compared with no additives ($p < 0.01$) (**A**). ### Difference compared with no additives ($p < 0.001$) (**B**).

On the other hand, both NPX photodegradation and the generation of NPX photoproducts induced by UV irradiation were suppressed completely in the presence of quercetin (molar ratio for NPX was 0.5, Figure 3D). Retention time of quercetin was ca. 23 min. The peaks of two NPX photoproducts were slightly detected but their peak area was less than the limit of quantification. The residual amount of NPX after UV irradiation (97.9 ± 0.7%) was significantly higher compared to that in the absence of quercetin

(84.1 ± 3.8%, Figure 4A). It is indicated that quercetin showed a photoprotective effect on NPX in the solid state. From the evaluation of the dose dependency of quercetin, NPX photodegradation and the generation of NPX photoproducts were suppressed partially even at a lower concentration (molar ratios for NPX was 0.125–0.25). Completely suppression was observed at a higher concentration (molar ratios for NPX was 0.5–1) as shown in Figure 4. From these results, it is suggested that quercetin might show the photoprotective effect on NPX by antioxidative potency, resulting in quenching the excitation of NPX and deactivating the hydroxyl radicals, or by UV filtering potency, resulting in disruption of UV irradiation for NPX, when powders of an NPX-quercetin mixture were UV-irradiated.

### 3.2. Comparison of the Photoprotective Effect of Selected Polyphenols and Antioxidants

Based on the results obtained from the evaluation of the photostabilization potency of quercetin, the comparative experiments of the photoprotective effects of selected polyphenols (catechin, chlorogenic acid, curcumin, daidzein, genistein, quercetin, resveratrol, rutin, and zingerone), and antioxidants (ascorbic acid and isoascorbic acid) were performed (Figure 2). Our previous study showed that ascorbic acid, isoascorbic acid, catechin, and curcumin showed photoprotective effects on NPX photodegradation in an aqueous media as the same as in quercetin [23]. So, we evaluated their photoprotective potencies of them and additional polyphenols for UV-irradiated NPX in the solid state. NPX and each polyphenol or antioxidant were mixed at a molar ratio of NPX 0.5 because this dose of quercetin completely suppressed NPX photodegradation. A comparison of the residual amounts of NPX after UV irradiation for 24 h in the absence or presence of selected additives is shown in Figure 5. Four additives, including ascorbic acid, zingerone, catechin, and isoascorbic acid, showed no protective effects on NPX photodegradation judged by the residual amounts of NPX in their presence. These values were almost the same as in that in the absence of additives. In the presence of these four additives, the residual amounts of NPX after UV irradiation for 24 h were significantly less than those of control samples. On the other hand, curcumin and resveratrol suppressed NPX photodegradation completely, the same as in quercetin. When quercetin, curcumin, and resveratrol were mixed with NPX, the residual amounts of NPX after UV irradiation (97.9 ± 0.7%, 98.9 ± 0.6% and 99.4 ± 1.1%, respectively) were significantly higher compared to that without additives (84.1 ± 3.8%), and comparable to those of their control samples (100.0 ± 0.3%, 99.2 ± 0.7% and 100.6 ± 1.2%, respectively). Other additives, including rutin, daidzein, chlorogenic acid, and genistein, showed photoprotective effects moderately on NPX photodegradation. The residual amounts of NPX after UV irradiation in the presence of rutin, daidzein, chlorogenic acid, and genistein were 91.5 ± 3.0%, 92.7 ± 1.1%, 94.1 ± 0.7%, and 97.3 ± 0.8%, respectively, which were significantly higher compared to that without additives, but these values were significantly less than those of their control samples (100.0 ± 0.4%, 99.0 ± 1.1%, 98.5 ± 0.7% and 100.7 ± 1.1%, respectively). These results indicated that some of the polyphenols tested in this study showed significant photoprotective effects on NPX photodegradation in the solid state. Quercetin, curcumin, and resveratrol showed remarkable protective potencies for NPX photodegradation.

In addition, quercetin and resveratrol showed high photostability because their residual amounts after UV irradiation were 98.0 ± 0.7% and 101.4 ± 1.3%, respectively (Figure S2), which were the same values as in those of control samples (98.5 ± 0.6% and 100.9 ± 1.1%, respectively). Daidzein, chlorogenic acid, and genistein were also photostable, but some polyphenols, including zingerone, catechin, rutin, and curcumin, were photodegradable. Several studies have been reported for the photodegdabilitiy of catechin and curcumin [26,27]. UV irradiation-induced the photodegradation of these polyphenols in addition to NPX. Furthermore, the residual amounts of curcumin, ascorbic acid, and isoascorbic acid in their control samples were 82.8 ± 1.2%, 91.8 ± 1.7%, and 93.2 ± 2.8%, respectively, indicating that these additives were degraded by the factors except for UV irradiation (probably oxidation, hygroscopicity and photodegradation by room-light irradiation during sample preparation, and so on). Based on results from the evaluations of both

NPX photostabilization and photostability of tested additives, it is tempting to speculate that quercetin and resveratrol might be good photostabilizers for powders and granules of photodegradable pharmaceuticals.

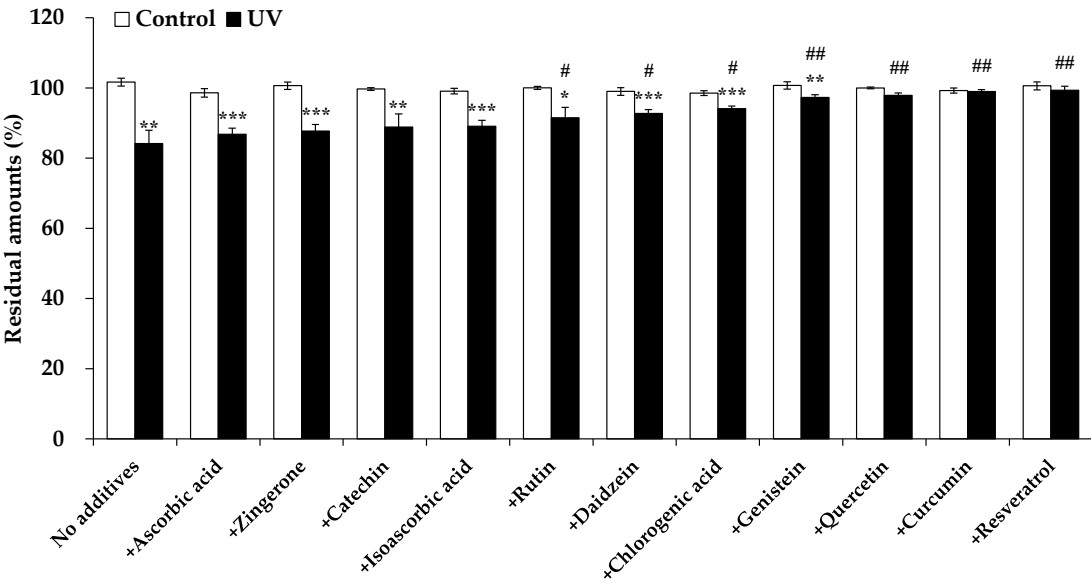

**Figure 5.** Photoprotective effects of selected additives on NPX photodegradation in a powder. Values represent mean ± S.D. (n = 4). * Difference compared with control ($p < 0.05$), **difference compared with control ($p < 0.01$), *** difference compared with control ($p < 0.001$), # difference compared with no additives ($p < 0.05$) and ## difference compared with no additives ($p < 0.01$).

### 3.3. Mechanism Elucidation of the Photoprotective Effects of Selected Additives

To clarify the mechanisms of the highly protective effects of quercetin, curcumin, and resveratrol on NPX photodegradation in the solid state, their antioxidative potencies were determined by the PAO test, as shown in Figure 6. From the results of the PAO test, there was no correlation between photoprotective effects and antioxidative potencies among selected polyphenols and antioxidants. Quercetin and resveratrol showed more significant antioxidative activities compared to the other tested compounds except for catechin and rutin. On the other hand, the antioxidative activity of curcumin was the same as in those of ascorbic acid and isoascorbic acid although curcumin showed a more photoprotective effect on NPX photodegradation compared to ascorbic acid and isoascorbic acid (Figure 5). In the case of catechin, its antioxidative activity was comparable to quercetin but their photoprotective effects on NPX were different (Figure 5). These results make it possible to confirm that an antioxidative potency has no contribution to the suppression of NPX photodegradation induced by UV irradiation in the solid state, and quercetin and resveratrol show highly protective effects by other natures. Interestingly, catechin showed a significant protective effect on NPX photodegradation in an aqueous media as shown in our previous study [23]. It is suggested that the status of photodegradable pharmaceuticals is an important factor to improve their photodegradability through the addition of photostabilizers.

Next, the UV absorption spectra of selected polyphenols and antioxidants were recorded to determine their absorption-maximum wavelengths ($\lambda_{max}$, nm) and molar absorption coefficients ($\varepsilon$, L mol$^{-1}$ cm$^{-1}$) as an indicator of UV filtering potency. NPX showed characteristic absorption in the wavelength above 260 nm although ascorbic acid and isoascorbic acid had no absorption as shown in Table 1. Photoprotective polyphenols for NPX showed more significant UV absorption compared to NPX due to their higher $\varepsilon$ values. Especially, quercetin, curcumin, and resveratrol had bigger $\varepsilon$ values around 260 nm and 450 nm, compared to the other tested compounds and NPX. Their $\lambda_{max}$ and $\varepsilon$ were as follows; quercetin 370 nm (23,971 L mol$^{-1}$ cm$^{-1}$), curcumin 424 nm and 263 nm

(72,442 L mol$^{-1}$ cm$^{-1}$ and 16,933 L mol$^{-1}$ cm$^{-1}$, respectively) and resveratrol 308 nm (34,140 L mol$^{-1}$ cm$^{-1}$). It is suggested that these three polyphenols might act as an efficient UV filter, which suppresses the excitation of NPX by disruption of UV irradiation, but poor photostabilizers such as ascorbic acid and isoascorbic acid might have no protective effects on account of their low UV filtering activities. It is proved that catechin had no photoprotective effect on NPX photodegradation due to showing insufficient UV absorption in the longer wavelength otherwise its higher antioxidative potencies.

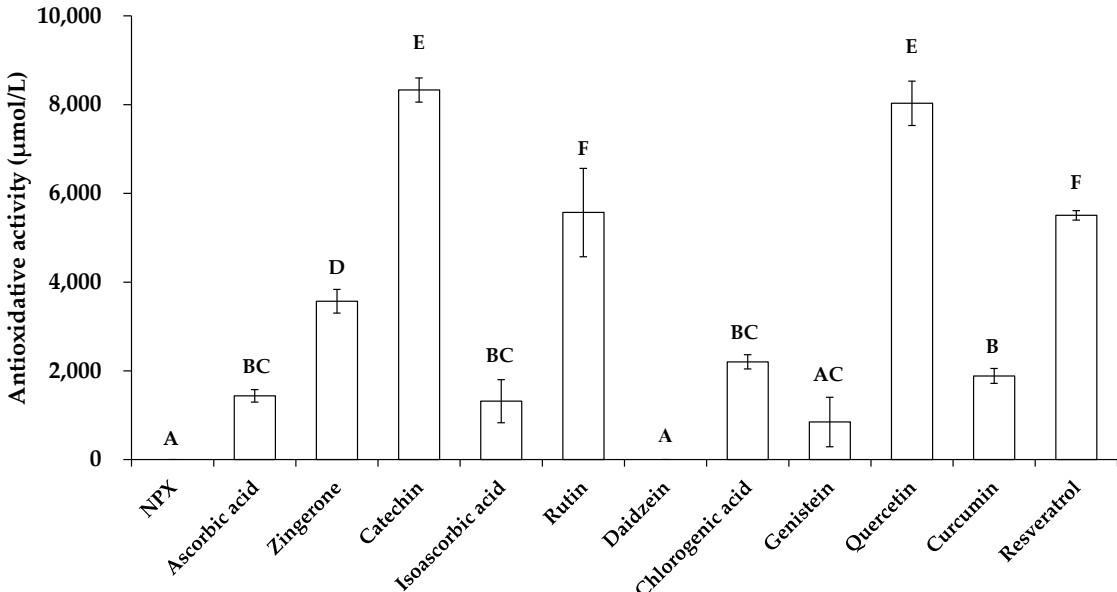

**Figure 6.** Antioxidative activities of NPX and selected additives. Values represent mean ± S.D. (n = 3). $^{A–F}$ Means without common superscripts are significantly different ($p < 0.05$).

**Table 1.** Absorption-maximum wavelengths ($\lambda_{max}$, nm) and molar absorbance coefficients ($\varepsilon$, L mol$^{-1}$ cm$^{-1}$) of NPX and tested additives were analyzed using UV/Vis spectrophotometer.

| | $\lambda_{max}$ (nm) and $\varepsilon$ (L mol$^{-1}$ cm$^{-1}$) in the Wavelength above 260 nm |
|---|---|
| NPX | 332 nm (1853), 318 nm (1481), 272 nm (5178), 263 nm (5144) |
| Ascorbic acid | - |
| Zingerone | 282 nm (2984) |
| Catechin | 280 nm (4114) |
| Isoascorbic acid | - |
| Rutin | 360 nm (17,607) |
| Daidzein | 303 nm (10,279) |
| Chlorogenic acid | 329 nm (20,032), 301 nm (15,213) |
| Genistein | 262 nm (44,251) |
| Quercetin | 370 nm (23,971) |
| Curcumin | 424 nm (72,442), 263 nm (16,933) |
| Resveratrol | 308 nm (34,140) |

From these results, it is clarified that a higher UV filtering activity is necessary for a better photostabilizer to photodegradable pharmaceuticals in the solid state. The summary of obtained results from photostabilization evaluation, determination of antioxidative potencies, and UV spectral analysis are shown in Table S1. Some of the selected additives suppressed NPX photodegradation due to their higher UV filtering activities, not antioxidative potencies. Quercetin, curcumin, and resveratrol completely suppressed NPX photodegradation on account of their remarkable UV absorption in the longer wavelength. Especially, quercetin and resveratrol disrupt the UV absorption of a compound, and they are photostable, suggesting that these polyphenols might suppress the photodegradation

and the generation of photoproducts for various photodegradable pharmaceuticals in the solid state, in addition to NPX. The photoprotective effect of curcumin was remarkable, but it might be weakened depending on photo-exposure time due to its lower stability. Several reports show the expression of toxicological potencies of photosensitive pharmaceuticals derived from the generation of photoproducts by UV irradiation [28–31], and the NPX photoproduct is known as toxicological potent in ecotoxicological tests [8,32]. In the case of NPX, it is proved indirectly that most selected polyphenols suppress the generation of NPX photoproduct partially or completely resulting in the reduction of ecotoxicological potencies induced by UV irradiation.

## 4. Conclusions

This work revealed that it is important for additives for the purpose of the photostabilization of NPX in the solid state to have sufficient UV filtering activity. In the absence of a UV absorptive additive, NPX in the solid state is excited by UV irradiation following to proceeding the photodegradation (Figure 7A). In contrast, in the presence of some UV absorptive substances such as quercetin, UV irradiation for NPX is disrupted resulting in the suppression of its photodegradation (Figure 7B). The addition of UV absorptive additives including selected polyphenols might be an efficient tool for the improvement of the photostability of photodegradable pharmaceuticals in the solid state. It might be a useful method to make a photostabilization for crushed or decapsulated medicines, of which APIs are photosensitive, in clinical situations. Additional research focused on the photoprotective effects of other additives is required for the development of a photostabilization strategy.

**A. In the absence of UV absorptive substances**

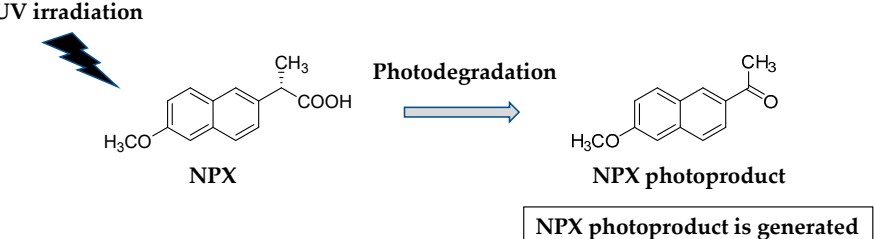

**B. In the presence of UV absorptive substances**

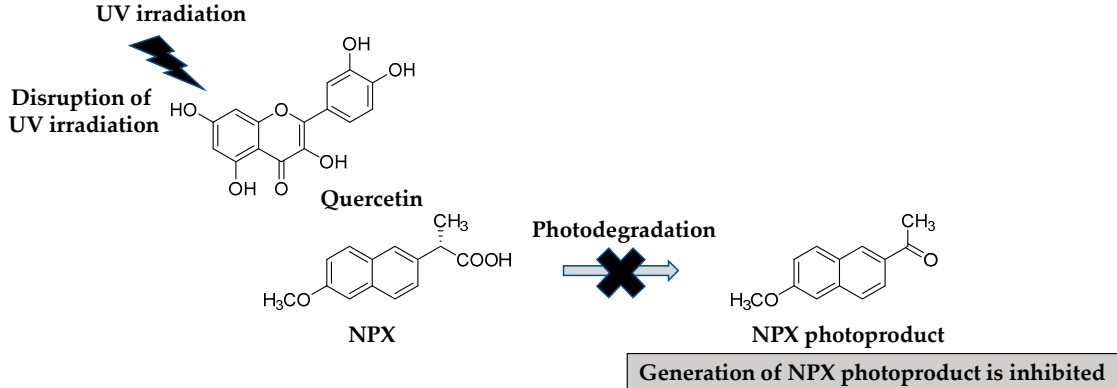

**Figure 7.** NPX is photodegraded by UV irradiation in the absence of UV absorptive substances (**A**) but they inhibit NPX photodegradation in a UV-irradiated powder by disruption of UV irradiation (**B**).

**Supplementary Materials:** The following supporting information can be downloaded at: https://www. mdpi.com/article/10.3390/photochem2040056/s1, Table S1: Summary of obtained results from photostabilization evaluation, determination of antioxidative potencies and UV spectral analysis; Figure S1: The scheme of experimental procedure of the evaluation of photostabilization potencies of selected additives for NPX photodegradation; Figure S2: Residual amounts of selected additives in the UV-irradiated powder in the presence of NPX. Values represent mean ± S.D. (n = 4). * Difference compared with control ($p < 0.05$) and *** difference compared with control ($p < 0.001$).

**Author Contributions:** Conceptualization, K.K. and H.N.; Methodology, K.K. and A.M.; Formal Analysis, K.K.; Investigation, K.K.; Resources, H.N.; Data Curation, K.K.; Writing—Original Draft Preparation, K.K.; Writing—Review & Editing, H.N.; Supervision, H.N.; Project Administration, K.K. and H.N.; Funding Acquisition, K.K. All authors have read and agreed to the published version of the manuscript.

**Funding:** This study was supported by JSPS KAKENHI Grant Number JP 20K15980.

**Conflicts of Interest:** The authors declare no conflict of interest.

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
