# Peer review of "Photoprotective Effects of Selected Polyphenols and Antioxidants on Naproxen Photodegradability in the Solid-State"

_2673-7256, doi:10.3390/photochem2040056_

Round 1

Reviewer 1 Report

The manuscript entitled “Photoprotective Effects of Selected Polyphenols and Antioxidants on Naproxen Photodegradability in the Sold-State” by Kawabata et al. describes the protective efficiency of several polyphenols and antioxidants towards Naproxen (an anti-inflammatory drug) when exposed to UV in its solid state. The work is well written and the data are well exposed; it was interesting to read and seems a good addition to the previous work of the authors. I however have some suggestions.

The authors should better explain that the idea, as I have understood it, is to find a material to shield the drug in solid state, for example as an encapsuling layer on a pill, against UV. Try and give a more visual description of the application.

Different graphs such as the one with Residual amount of NPX and Generation rates of NPX photoproduct in Fig.3 would be better set apart from the reader if they had a different graphical layout (colors for example). The title of Fig.3B should be revised since it says “photoprpduct” instead of “photoproduct”.

If the authors could find a way to make it clearer to the reader that the photoprotection occurs for the UV absorption rather than for the antioxidative power it would help, possibly a scheme or table that shows the numbers for photoprotection, antioxidant efficiency and UV absorption for some selected molecules.

Author Response

Response to Reviewer 1 Comments

Photochem-2015760

Kawabata, K., Miyoshi, A., Nishi, H. Photoprotective Effects of Selected Polyphenols and Antioxidants on Naproxen Photodegradability in the Solid-State.

Thank you for your letter and the reviewer’s comments concerning our manuscript. We have studied their comments carefully and have made following revisions which we hope meet their approval.

(Point 1)

The authors should better explain that the idea, as I have understood it, is to find a material to shield the drug in solid state, for example as an encapsuling layer on a pill, against UV. Try and give a more visual description of the application.

(Response 1)

The purpose of our study is to develop a useful method to make a photostabilization by addition of some compounds for crushed- or decapsuled medicines, of which APIs are photosensitive, in clinical situations. So, the photoprotective effects of additives were not evaluated to discover some materials as photostabilizers in formulations. Please accept an our concept for a photostabilization strategy.

(Point 2)

Different graphs such as the one with Residual amount of NPX and Generation rates of NPX photoproduct in Fig.3 would be better set apart from the reader if they had a different graphical layout (colors for example). The title of Fig.3B should be revised since it says “photoprpduct” instead of “photoproduct”.

(Response 2)

Thank you for your suggestion for design of Figure 4B. The colors of columns of Figure 4B is changed. Also, the title of Figure 4B is modified as “Generation rates of NPX photoproduct”.

(Point 3)

If the authors could find a way to make it clearer to the reader that the photoprotection occurs for the UV absorption rather than for the antioxidative power it would help, possibly a scheme or table that shows the numbers for photoprotection, antioxidant efficiency and UV absorption for some selected molecules.

(Response 3)

We appreciate for your precious comment. Table S1, which describe the summary of obtained results of photostabilization evaluation, determination of antioxidative potencies and UV spectral analysis, is added in supporting information.

We hope that revised manuscript is now acceptable for publication.

Best regards.

Your Sincerely,

NAME: Kohei Kawabata, Ph. D.

ADRESS: Faculty of Pharmacy, Yasuda Women’s University, 6-13-1 Yasuhigashi, Asaminami-ku, Hirosshima 731-0153, Japan

EMAIL: kawabata-k@yasuda-u.ac.jp

TEL: 81-82-878-9440

FAX: 81-82-878-9540

Reviewer 2 Report

The manuscript ID: photochem-2015760 “Photoprotective Effects of Selected Polyphenols and Antioxidants on Naproxen Photodegradability in the Solid-State”. In this article the author examined the protective effects of selected polyphenols and antioxidants on NPX photodegradation in the solid-state were evaluated. The photo stabilization of naproxen (NPX) in the solid-state to have a sufficient UV filtering activity and NPX is excited by UV irradiation following to proceeding the photodegradation. The addition of UV absorptive additives including selected polyphenols might be an efficient tool for the improvement of the photostability of photodegradable pharmaceuticals in the solid-state. The author most of the paper was well-organized and presented results.  However, some issues still need to be addressed before publication in Photochem. The topic is interesting for the referee. Therefore, I recommend publication only after minor revisions.

 1) I think that Figure S1, needs to be moved to the main paper from supporting information.

 2) Page No:3; Line No: 103-104: the author has written “Control samples were prepared using the same procedures but covered with aluminum foil to interrupt the photo-exposure”. The author needs to include the experiment picture/image in supporting information for better understanding.

 3) Page No:4; Figure 2; In figure D. NPX + Quercetin, UV, the figure is showing two small peaks at around 12.5 and 18.5. the author needs to explain these peaks.

 4) Page No:5; Figure 3; B. Generation rates of NPX photoprpduct, the author needs to correct/include the correct form of “photoprpduct”.

 5) Page No: 8; Line 272-285: UV absorption spectra, the author has provided only data. I think that the author needs to include the UV absorption spectra of all the compounds. Which solvent has been used for recording UV absorption spectra?

Author Response

Response to Reviewer 2 Comments

Photochem-2015760

Kawabata, K., Miyoshi, A., Nishi, H. Photoprotective Effects of Selected Polyphenols and Antioxidants on Naproxen Photodegradability in the Solid-State.

Thank you for your letter and the reviewer’s comments concerning our manuscript. We have studied their comments carefully and have made following revisions which we hope meet their approval.

(Point 1)

I think that Figure S1, needs to be moved to the main paper from supporting information.

(Response 1)

Figure S1 is assigned as Figure 2 in revised manuscript.

(Point 2)

Page No:3; Line No: 103-104: the author has written “Control samples were prepared using the same procedures but covered with aluminum foil to interrupt the photo-exposure”. The author needs to include the experiment picture/image in supporting information for better understanding.

(Response 2)

The scheme of experimental procedure of the evaluation of photostabilization potencies of selected additives for NPX photodegradation is added in supporting information as Figure S1.

(Point 3)

Page No:4; Figure 2; In figure D. NPX + Quercetin, UV, the figure is showing two small peaks at around 12.5 and 18.5. the author needs to explain these peaks.

(Response 3)

The explanation for two small peaks of NPX photoproducts in Figure 3D is added in line No. 191-192 in page No. 6.

(Point 4)

Page No:5; Figure 3; B. Generation rates of NPX photoprpduct, the author needs to correct/include the correct form of “photoprpduct”.

(Response 4)

Figure title of Figure 4B is modified as “Generation rates of NPX photoproduct”.

(Point 5)

Page No: 8; Line 272-285: UV absorption spectra, the author has provided only data. I think that the author needs to include the UV absorption spectra of all the compounds. Which solvent has been used for recording UV absorption spectra?

(Response 5)

Thank you for your sincerely suggestion of addition of UV absorption spectra of test additives. We tried to include them as figure firstly, but it was impossible due to that it was difficult to export obtained UV absorption spectra as word files and PDF files. The spec of PC which is used as system controller is poor, so we decided to summarize the results of UV spectral analysis as Table 1. Also, it is difficult to carry out UV spectral analysis again because revised manuscript must be submitted within 5 days. So we are sorry for not accepting your comments.

We hope that revised manuscript is now acceptable for publication.

Best regards.

Your Sincerely,

NAME: Kohei Kawabata, Ph. D.

ADRESS: Faculty of Pharmacy, Yasuda Women’s University, 6-13-1 Yasuhigashi, Asaminami-ku, Hirosshima 731-0153, Japan

EMAIL: kawabata-k@yasuda-u.ac.jp

TEL: 81-82-878-9440

FAX: 81-82-878-9540